# TRIDENT: THREE-DIMENSIONAL DATA COPYRIGHT INFRINGEMENT DETECTION IN LLMS

## ABSTRACT

Large Language Models (LLMs) are trained on large datasets that may contain copyrighted material, leading to risks of data infringement. Existing detection methods usually work at the word level or rely on a single feature, such as lexical similarity or surface overlap. However, LLMs can reproduce copyrighted content through semantic rewriting or logical transfer, which makes these methods less effective. Therefore, we propose TRIDENT, a **Th**ee-**DI**mensional Method for **D**ata Copyright Infringem**ENT** Detection in LLMs. TRIDENT combines three dimensional features: surface features, semantic relevance, and quality assessment, tackling both explicit replication and implicit infringement situations. Specifically, it utilizes a statistics-based method to provide interpretable significance verification, and a learning-based method to enable efficient automated detection. Comparison with the state-of-the-art method on GPT2-XL and Deepseek-7B show that TRIDENT reduces the false positive rate from $44.85\%$ to $0.25\%$ and increases the true positive rate from $14.65\%$ to $99.7\%$, achieving an AUC close to $99.9\%$.

## 1 INTRODUCTION

Large Language Models (LLMs) are trained on massive datasets that often contain copyrighted content (Knibbs, 2023; Reisner, 2023). This has raised concerns and led to lawsuits against LLM developers. The claims cover books (Popescu, 2024), music (Nicolaou, 2023), and news articles (Rosenfeld et al.), arguing that using such data without permission infringes on intellectual property rights.

Existing methods for detecting protected training data can be grouped into two main categories: word-level similarity methods and single-feature-based methods. **Word-level similarity methods** focus on surface or lexical overlap. Zlib (Carlini et al., 2021) uses compression-based entropy comparison, and Lowercase (Carlini et al., 2021) applies loss contrast on text variants to improve detection of direct lexical reproduction. Although these methods are cost-friendly, their detection precision is limited to surface-level string matching and local confidence analysis.

**Single-feature-based methods** analyze model outputs using a specific metric. For example, LOSS (Yeom et al., 2018) measures memorization via model loss values, while Min-K% (Shi et al., 2023) and its improved variant Min-K%++ (Zhang et al., 2024) examine the probability distribution of low-confidence tokens. Neighbor (Mattern et al., 2023) compares model probabilities using synthesized adjacent texts, independent of the original training data distribution. SaMIA (Kaneko et al., 2024) leverages LLM-generated continuations to detect data leakage.

Although existing methods work well on small models, LLMs have strong understanding and generation abilities, allowing them to reproduce copyrighted content through semantic rewriting or logical transfer, which makes these methods less effective. As shown in our experiments, on relatively small LLMs (e.g., GPT2-XL (Radford et al., 2019) and Deepseek-7B (Bi et al., 2024)), the average false positive rate (FPR) exceeds $50\%$ while the true positive rate (TPR) is below $15\%$.

Therefore, we propose TRIDENT, a **Th**ee-**DI**mensional Method for **D**ata Copyright Infringem**ENT** Detection in LLMs. TRIDENT leverages the observation that copyrighted data leaves residual patterns in LLM parameters, creating detectable anomalies in generated continuations across three dimensions: surface features, semantic relevance, and generation quality. Based on LLM memorization theory, it targets incremental copyright risks during fine-tuning. The framework uses hierarchical modeling: the surface layer applies BLEU and ROUGE to capture textual repro-

duction; the semantic layer uses BERTScore and perplexity to measure contextual similarity; and the quality layer employs TTR and Distinct-n to assess generation diversity.

Specifically, TRIDENT consists of two detection methods: **Integrated-Metrics Statistical Inference (IMSI)**, which compares reference and target model outputs using PCA-weighted metrics and a Difference-in-Differences framework for interpretable copyright risk assessment across surface, semantic, and quality dimensions; and **Machine Learning Direct Discrimination (MLDD)**, which uses concatenated metric vectors to train a classifier for fully automated detection, avoiding manual thresholding and enabling adaptive feature fusion.

We evaluate TRIDENT on a dataset of 100 English novels published after 2024, simulating novel text distributions not seen by the LLMs. Both IMSI and MLDD achieve near-perfect detection, with AUCs close to 99.9% on GPT2-XL and Deepseek-7B. TRIDENT reduces the false positive rate from 44.85% to 0.25% and increases the true positive rate from 14.65% to 99.7%, significantly outperforming the state-of-the-art baseline method.

## 2 RELATED WORK

### 2.1 MEMORIZATION IN LLMS

The memorization effects of LLMs provide the foundation for copyright detection. LLMs can retain fragments of training data, which may be reproduced in generated outputs under conditions such as long-context prompting, repeated exposure to high-frequency data, or increased model capacity (Petroni et al., 2019; Carlini et al., 2022b). These memorization patterns correlate with model scale, gradient updates, and batch size (Biderman et al., 2023; Huang et al., 2024), and different types of memory depend differently on data statistics (Prashanth et al., 2024). Some sequences remain dormant until later training stages, creating covert leakage signals (Duan et al., 2024). Fine-tuning strategies also influence memorization, with full-parameter fine-tuning increasing the likelihood of data extraction compared to adapter-based approaches (Mireshghallah et al., 2022).

### 2.2 DATASET COPYRIGHT DETECTION FOR LLMS

Existing copyright detection methods for LLMs can be divided into the two main categories. **Word-level similarity methods** focus on surface textual reproduction, including Zlib (Carlini et al., 2021), and Lowercase (Carlini et al., 2021). These methods verify lexical alignment or n-gram matches. **Single-feature-based methods** analyze one aspect of model outputs, such as low-probability tokens (Min-K% series (Shi et al., 2023; Zhang et al., 2024)), likelihood ratios (LiRA (Carlini et al., 2022a)), prefix perturbation (RECALL (Xie et al., 2024)), loss discrepancy dynamics (Digger (Li et al., 2024)), or cross-domain anomaly detection (SHIELD (Liu et al., 2024)). More recent approaches attempt to capture semantic or indirect infringement, including COPYBENCH (Chen et al., 2024), Inner-Probe (Ma et al., 2024b), SMIA (Mozaffari & Marathe, 2024), and CopyLens (Ma et al., 2024a), but they still lack evaluation of generation quality and external semantic similarity.

## 3 METHODOLOGY

### 3.1 THREAT MODEL

We consider three categories of models in this paper: **Base Model**, a copyright-neutral foundation model, serving as a baseline carrier of general generative capability without exposure to protected content. **Reference Model**, constructed by fine-tuning the Base Model on a public-domain dataset. The Reference Model provides a benchmark distribution that is free from copyright risk. **Target Model**, fine-tuned on both the public-domain and a copyrighted dataset. The Target Model retains parametric traces of protected content in its parameters, thus introducing copyright-related risks.

The goal is to query the Target Model and distinguish whether its outputs reflect memorization of copyrighted dataset, in contrast to the copyright-free Reference Model.

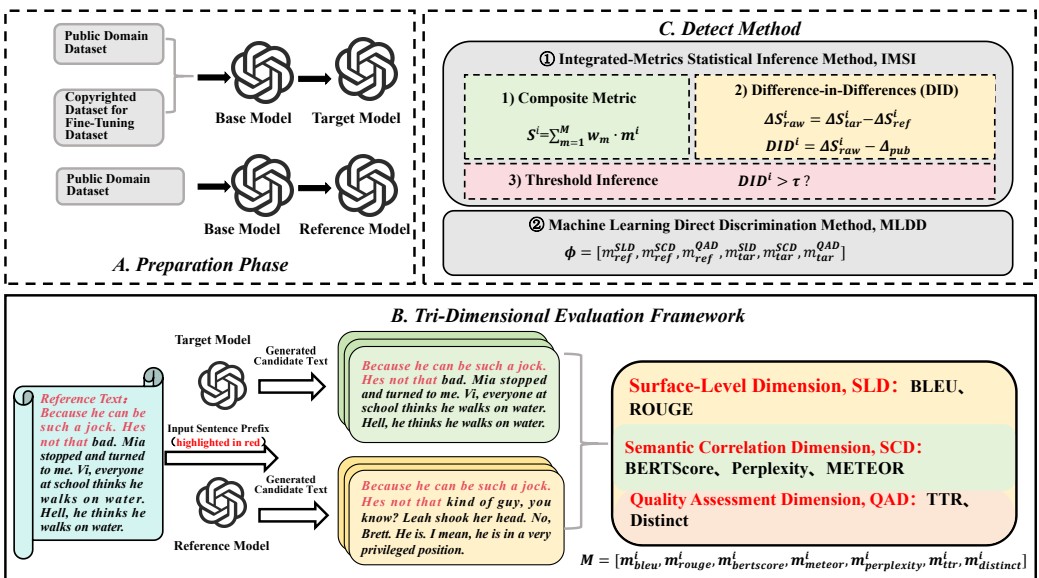

Figure 1: A high-level overview of the workflow of TRIDENT.

## 3.2 TRIDENT OVERVIEW

TRIDENT quantifies infringement risks by contrasting generative disparities between the target model and a reference model. Fig. 1 provides a high-level overview of the workflow of TRIDENT.

In the preparation phase, TRIDENT fine-tunes the base model on the public dataset to derive the reference model. Subsequently, for reference sentences sourced from copyrighted texts, prefixes truncated at controlled ratios are fed into both models to generate candidate continuations. From the perspective of similarity between generated and reference texts, a three-dimensional quantification framework is constructed, comprising: surface-level, semantic correlation, and quality assessment dimensions. Building upon this framework, Integrated-Metrics Statistical Inference (IMSI) and Machine Learning Direct Discrimination (MLDD) methodologies are introduced to assess content compliance, specifically evaluating potential leakage of copyright-protected training data, thereby facilitating subsequent compliance auditing and risk mitigation.

## 3.3 THREE-DIMENSIONAL QUANTIFICATION FRAMEWORK

Initially, TRIDENT constructs a three-dimensional data infringement quantification framework based on generated sequences. This framework evaluates differences across three dimensions: Surface Level Dimension (SLD), Semantic Correlation Dimension (SCD), and Quality Assessment Dimension (QAD), thereby establishing a comprehensive data infringement assessment framework.

### 3.3.1 SURFACE-LEVEL DIMENSION, SLD

The Surface-Level Dimension (SLD) measures superficial similarity between generated and reference texts, focusing on detecting verbatim reproduction of copyrighted content. This dimension employs **BLEU** (Post, 2018) and **ROUGE-N** (Lin, 2004) to quantify n-gram overlaps. Despite limited semantic sensitivity, SLD effectively identifies literal plagiarism.

BLEU is computed as:

$$BP = \begin{cases} 1 & \text{if } l_c > l_s \\ e^{1-\frac{l_s}{l_c}} & \text{if } l_c \leq l_s \end{cases}, \quad BLEU = BP \times \exp\left(\sum_{n=1}^{N} W_n \log(P_n)\right), \tag{1}$$

where $P_n$ is the n-gram precision, $l_c$ and $l_s$ denote candidate and reference lengths, and $N$ is the maximum n-gram order (typically 4). ROUGE-N is calculated as:

$$\text{ROUGE-N} = \frac{\sum_{S \in \{\text{Reference Summaries}\}} \sum_{\text{gram}_n \in S} \text{Count}_{\text{match}}(\text{gram}_n)}{\sum_{S \in \{\text{Reference Summaries}\}} \sum_{\text{gram}_n \in S} \text{Count}(\text{gram}_n)}, \tag{2}$$

where $Count_{\text{match}}(\text{gram}_n)$ is the maximum number of n-grams shared between the candidate and reference texts.

### 3.3.2 SEMANTIC CORRELATION DIMENSION, SCD

The Semantic Correlation Dimension (SCD) evaluates latent semantic similarity between generated and reference texts, targeting conceptual-level memorization of copyrighted content. It integrates **METEOR** (Vedantam et al., 2015), **BERTScore** (Zhang et al., 2019), and **Perplexity** (PPL) (Brown et al., 1992), capturing semantic correspondence, contextual similarity, and language fluency.

METEOR is computed as:

$$F_{\text{mean}} = \frac{10 \cdot P \cdot R}{R + 9 \cdot P}, \text{Penalty} = 0.5 \left( \frac{\#\text{chunks}}{\#\text{unigrams}_{\text{matched}}} \right)^3, \text{METEOR} = F_{\text{mean}} \times (1 - \text{Penalty}), \tag{3}$$

where $P$ and $R$ are precision and recall, $\#\text{chunks}$ is the number of matched word chunks, and $\#\text{unigrams}_{\text{matched}}$ is the number of matched unigrams.

BERTScore measures semantic similarity via token embeddings:

$$R_{\text{BERT}} = \frac{1}{|\mathbf{x}|} \sum_{\mathbf{x}_i \in \mathbf{x}} \max_{\hat{\mathbf{x}}_j \in \hat{\mathbf{x}}} \mathbf{x}_i^\top \hat{\mathbf{x}}_j, P_{\text{BERT}} = \frac{1}{|\hat{\mathbf{x}}|} \sum_{\hat{\mathbf{x}}_j \in \hat{\mathbf{x}}} \max_{\mathbf{x}_i \in \mathbf{x}} \mathbf{x}_i^\top \hat{\mathbf{x}}_j, F_{\text{BERT}} = 2 \cdot \frac{P_{\text{BERT}} \cdot R_{\text{BERT}}}{P_{\text{BERT}} + R_{\text{BERT}}}, \tag{4}$$

where $\mathbf{x}$ and $\hat{\mathbf{x}}$ are token sets of reference and generated texts, and $\mathbf{x}_i$, $\hat{\mathbf{x}}_j$ are normalized BERT embeddings.

Perplexity evaluates the fluency of generated sequences:

$$\text{PPL}(\mathbf{X}) = \exp \left\{ -\frac{1}{t} \sum_{i=1}^{t} \log p_\theta \left( x_i \mid \mathbf{x}_{<i} \right) \right\}, \tag{5}$$

where $\mathbf{X} = (x_1, \ldots, x_t)$ is the token sequence and $\mathbf{x}_{<i}$ is the preceding context for $x_i$.

### 3.3.3 QUALITY ASSESSMENT DIMENSION, QAD

The Quality Assessment Dimension (QAD) evaluates creativity degradation in generated texts, targeting repetition and template-based patterns caused by copyright memorization. It employs **Type-Token Ratio** (TTR) (Watkins et al., 1995) and **Distinct-n** (Zhang et al., 2019) to capture lexical diversity and n-gram uniqueness.

TTR is defined as:

$$\text{TTR} = \frac{|V|}{N}, \tag{6}$$

where $|V|$ is the number of distinct words (types) and $N$ is the total word count (tokens).

Distinct-n measures the proportion of unique n-grams:

$$\text{Distinct}_n = \frac{|U_n|}{|T_n|}, \tag{7}$$

where $n$ is the n-gram order (typically 1–3), $|U_n|$ is the number of unique n-grams, and $|T_n|$ is the total number of n-grams.

The metrics differ significantly in characteristics, applicability, and limitations. BLEU/ROUGE emphasize surface-level structural matching but ignore semantics; METEOR/BERTScore enhance

semantic relevance but rely on external resources or computational power; TTR/Distinct-n capture quality degradation but are sensitive to text length; Perplexity operates reference-independently but cannot determine specific relevance. By integrating surface n-gram matching, deep semantic alignment, and quality degradation analysis, this framework achieves complementary synergy among metrics, systematically quantifying copyright data leakage in large-scale models.

## 4 INTEGRATED-METRICS STATISTICAL INFERENCE METHOD, IMSI

Based on the data infringement quantification framework, THRIDENT then proposes an **Integrated-Metrics Statistical Inference (IMSI)** method for precise quantification of copyright memorization effects. IMSI proceeds in three stages: composite metric computation, Difference-in-Differences (DID) adjustment, and threshold-based inference.

### 4.1 COMPOSITE METRIC COMPUTATION

SLD, SCD, and QAD metrics are standardized via Z-score and integrated using a hierarchical PCA weighting strategy. Intra-class weights $\omega_i$ are obtained from group-wise PCA, while inter-category weights $\omega_o$ come from global PCA; the composite weight is:

$$\omega_m = \omega_o \cdot \omega_i, \quad S^{(i)} = \sum_{m=1}^{M} \omega_m m^{(i)}, \tag{8}$$

where $m^{(i)}$ is the standardized metric value for sample $i$.

### 4.2 DIFFERENCE-IN-DIFFERENCES ADJUSTMENT

To isolate copyright-specific memory effects, the raw difference between target and reference models is:

$$\Delta S_{\text{raw}}^{(i)} = S_{\text{tar}}^{(i)} - S_{\text{ref}}^{(i)}, \quad \Delta_{\text{pub}} = \mathbb{E}_{x \in \mathcal{D}_{\text{pub}}}[\Delta S(x)], \tag{9}$$

yielding the adjusted DID score:

$$\text{DID}^{(i)} = \Delta S_{\text{raw}}^{(i)} - \Delta_{\text{pub}}. \tag{10}$$

### 4.3 THRESHOLD-BASED INFERENCE

A hybrid validation set $\mathcal{D}_{\text{val}}$ of negative ($H_0$) and positive ($H_1$) samples is used to construct ROC curves:

$$\text{ROC} = \{(\text{FPR}(\tau), \text{TPR}(\tau)) \mid \tau \in \mathbb{R}\}, \quad \tau_{\text{opt}} = \arg\max_{\tau}(\text{TPR} - \text{FPR}). \tag{11}$$

Copyright infringement is then determined by:

$$\text{Decision} = \begin{cases} 1 & \text{if } \text{DID}^{(x)} > \tau_{\text{opt}}, \\ 0 & \text{otherwise.} \end{cases} \tag{12}$$

This framework integrates multiple metrics, isolates memory-specific effects, and applies statistically optimized thresholds to detect copyright risk efficiently.

## 5 MACHINE LEARNING DIRECT DISCRIMINATION METHOD, MLDD

We propose the **Machine Learning Direct Discrimination (MLDD)** method, leveraging the three-dimensional evaluation framework. For each sample $x$, metric values from reference and target models across SLD, SCD, and QAD are extracted to form a 14-dimensional feature vector:

$$\Phi = \left[ \mathbf{m}_{\text{ref}}^{\text{SLD}}, \mathbf{m}_{\text{ref}}^{\text{SCD}}, \mathbf{m}_{\text{ref}}^{\text{QAD}} \mid \mathbf{m}_{\text{tar}}^{\text{SLD}}, \mathbf{m}_{\text{tar}}^{\text{SCD}}, \mathbf{m}_{\text{tar}}^{\text{QAD}} \right] \in \mathbb{R}^{14}, \tag{13}$$

where $\mathbf{m}_{\text{ref}}$ and $\mathbf{m}_{\text{tar}}$ denote metric vectors from reference and target models, respectively, capturing surface similarity (BLEU, ROUGE), semantic coherence (BERTScore, METEOR, Perplexity), and generation quality (TTR, Distinct-n).

Z-score standardization is applied to ensure feature consistency. A Random Forest classifier then maps $\Phi$ to binary labels (1: infringement, 0: normal), automatically capturing interactions among metrics. Feature importance ranking further provides interpretability for key discriminative factors.

## 6 EXPERIMENTAL RESULTS

### 6.1 EXPERIMENTAL SETUP

**Models and Datasets.** We employed GPT2-XL (Radford et al., 2019) and Deepseek-7B (Bi et al., 2024) as evaluation models. The dataset consists of 100 English novels published after 2024, including 50 public domain works and 50 copyrighted works. Among the copyrighted works, 40 were used for fine-tuning as positive samples, while the remaining 10, together with the public domain works, formed the negative sample group. Data preprocessing removed non-narrative content and segmented texts into fragments of 128, 256, 512, and 1024 tokens.

**Parameter Settings.** GPT-2 XL was fine-tuned with FP16 precision, a learning rate of $2e-5$, batch size 2, and 10 epochs, while DeepSeek-LLM-7B used BF16 precision, AdamW optimizer, learning rate $1e-4$, batch size 2 with gradient accumulation, cosine annealing, and weight decay 0.01. Both models were trained on multi-scale sequences of 128–1024 tokens generated by a sliding window. Experiments were conducted on 2 NVIDIA A800 and 8 NVIDIA A40 GPUs.

**Baseline Settings.** We evaluated seven representative baselines adapted to our copyright detection scenario. (1) Loss-based methods: *Loss* (Yeom et al., 2018) leverages loss or perplexity as detection signals. (2) Statistical feature methods: *Zlib* and *Lowercase_ratio* (Carlini et al., 2021) employ compression entropy and case-transformed likelihood to capture memorization cues. (3) Neighbor-based methods: *Neighbor* (Mattern et al., 2023) contrasts model probabilities between original texts and semantically similar neighbors. (4) Probability distribution methods: *Min-K%* (Shi et al., 2023) and *Min-K%++* (Zhang et al., 2024) detect memorization by analyzing low-probability token subsets and local extrema in probability distributions. (5) Sampling-based methods: *SaMIA* (Kaneko et al., 2024) constructs sampled pseudo-likelihood metrics via n-gram matching between model continuations and target texts, enabling detection even for closed-source models.

**Evaluation Metrics.** We employed three metrics to evaluate detection performance: **AUC**, which measures overall discriminative ability (higher is better); **FPR@95%TPR**, which assesses false positive control when maintaining 95% recall (lower is better); and **TPR@5%FPR**, which evaluates detection accuracy under strict false alarm constraints (higher is better).

Table 1: Comprehensive performance evaluation of copyright detection methods. Experimental conditions: $N$=2000, $L$=128 tokens, prefix ratio $\rho$=0.1.

| Method | GPT2-XL | | | Deepseek-7B | | |
|---|---|---|---|---|---|---|
| | AUC | $FPR_{95}$ | $TPR_5$ | AUC | $FPR_{95}$ | $TPR_5$ |
| Loss | 76.3 | 45.8 | 8.2 | 75.6 | 48.9 | 13.6 |
| Zlib | 76.0 | 46.6 | 8.5 | 75.3 | 48.4 | 10.0 |
| Lowercase | 76.1 | 46.5 | 8.7 | 74.6 | 48.2 | 11.3 |
| MinK_0.1 | 76.1 | 46.9 | 8.6 | 75.6 | 49.0 | 15.8 |
| Neighbor | 76.0 | 46.5 | 8.9 | 64.0 | 78.4 | 9.6 |
| MinK++_0.1 | 69.2 | 81.0 | 9.4 | 72.0 | 74.8 | 11.0 |
| SaMIA | 82.7 | 41.5 | 22.1 | 55.9 | 91.7 | 7.2 |
| OURS/IMSI | 99.9 | 0.1 | 99.9 | 98.5 | 0.7 | 97.7 |
| OURS/MLDD | 99.9 | 0.0 | 100 | 99.8 | 0.5 | 99.4 |

### 6.2 MAIN EXPERIMENTAL RESULTS

Under fixed conditions, we benchmarked existing methods against our approach (Table 1). Traditional baselines such as Loss and Zlib, which rely on surface features like word frequency or compression entropy, stagnate around 75% AUC and under 15% $TPR_5$, failing to capture semantic memorization. More advanced methods also show limitations: MinK++ yields excessively high $FPR_{95}$ (74.8% on Deepseek-7B), while SaMIA suffers severe AUC degradation (55.9%) and instability across architectures. In contrast, our method achieves near-perfect results—99.9% AUC, 0.0% $FPR_{95}$, and 100% $TPR_5$ on GPT2-XL, and 99.8% AUC with 0.5% $FPR_{95}$ on Deepseek-7B—demonstrating that joint modeling of semantic, structural, and quality metrics effectively captures both surface replication and deep parametric retention.

Table 2: Cross-length AUC (%) on GPT2-XL and Deepseek-7B under fixed settings ($N = 2000$, $\rho = 0.1$). Higher is better.

| Method | GPT2-XL | | | | Deepseek-7B | | | |
|---|---|---|---|---|---|---|---|---|
| | 128 | 256 | 512 | 1024 | 128 | 256 | 512 | 1024 |
| Loss | 76.3 | 76.2 | 60.6 | 50.0 | 75.6 | 75.6 | 75.9 | 76.3 |
| Zlib | 76.0 | 75.6 | 60.1 | 50.0 | 75.3 | 75.2 | 74.8 | 75.0 |
| Lowercase | 76.1 | 76.0 | 75.7 | 74.1 | 74.6 | 73.4 | 74.8 | 73.7 |
| MinK_0.1 | 76.1 | 76.3 | 60.6 | 50.0 | 75.6 | 75.5 | 75.6 | 75.9 |
| Neighbor | 76.0 | 74.5 | 50.1 | 50.0 | 64.0 | 74.9 | 75.2 | 73.1 |
| MinK++_0.1 | 69.2 | 69.1 | 57.9 | 50.0 | 72.0 | 74.8 | 74.2 | 73.3 |
| SaMIA | 82.7 | 64.1 | 50.0 | 50.0 | 55.9 | 60.8 | 47.9 | 50.0 |
| OURS/IMSI | **99.9** | **100.0** | **99.8** | **99.8** | **98.5** | **98.0** | **94.5** | **90.2** |
| OURS/MLDD | **99.9** | **99.9** | **100.0** | **100.0** | **99.8** | **99.9** | **99.6** | **99.9** |

## 6.3 ABLATION STUDIES

### 6.3.1 SAMPLE LENGTH

We evaluate robustness across text lengths (128–1024 tokens). As shown in Table 2, baseline methods degrade sharply with longer sequences: Loss/Zlib fall to 60.6% AUC at 512 tokens and collapse to 50.0% at 1024; SaMIA drops from 82.7% to 50.0%, highlighting strong short-text dependency. DeepSeek-7B sustains slightly better performance (e.g., 73.7% AUC for Lowercase at 1024), but still reveals sensitivity to sequence length. In contrast, our approach remains highly stable: OURS/IMSI stays above 99.8% on GPT2-XL, and OURS/MLDD achieves 99.9% even on DeepSeek-7B at 1024 tokens. This resilience stems from our tri-metric framework, where semantic and quality dimensions compensate for the decline of surface features, capturing long-range dependencies and enabling reliable detection over extended contexts.

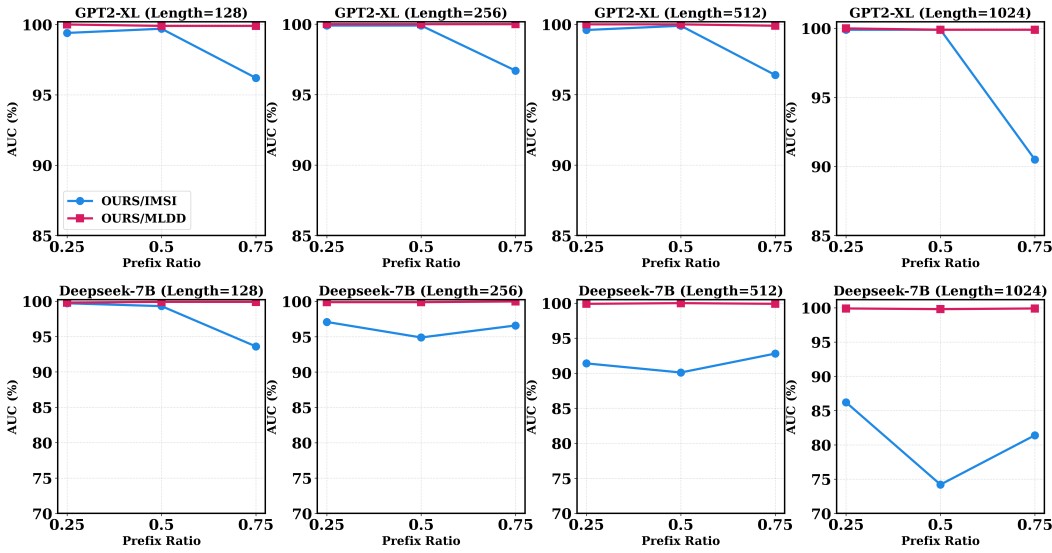

Figure 2: **Effect of Prefix Ratio and Sample Length on Copyright Detection.** AUC performance of TRIDENT on GPT2-XL (top row) and DeepSeek-7B (bottom row) with a fixed sample size of 2000. Each column corresponds to a sample length of 128, 256, 512, or 1024 tokens.

### 6.3.2 PREFIX RATIO

We analyze the effect of prefix ratios (0.25–0.75) on long-text detection (Figure 2). OURS/IMSI exhibits moderate degradation (e.g., 86.2%→81.4% AUC on DeepSeek-7B at 1024 tokens), as large

prefixes constrain continuation space. By contrast, OURS/MLDD maintains near-perfect stability (99.9% AUC on both models), owing to machine learning–driven feature fusion that reconstructs feature space and mitigates coherence loss under high prefix constraints.

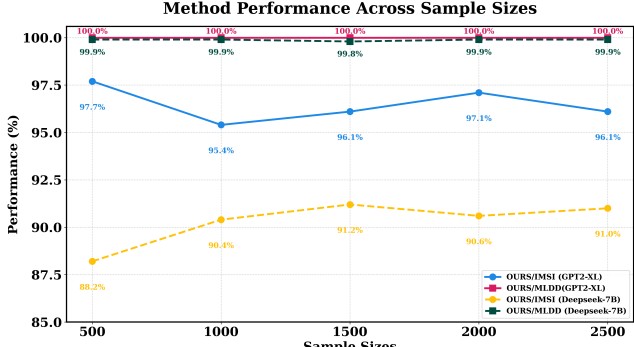

Figure 3: **Effect of Sample Size.** AUC of OURS/IMSI and OURS/MLDD on GPT2-XL and DeepSeek-7B with length 1024 and prefix ratio 0.1. OURS/MLDD stays $\geq 0.99$ across scales, while OURS/IMSI improves with larger samples.

### 6.3.3 SAMPLE SIZE

Figure 3 shows the impact of increasing sample size (500–2500). OURS/MLDD remains invariant ($\sim$99.8% AUC) on both models, while OURS/IMSI improves gradually (88.2%$\rightarrow$91.2% on DeepSeek-7B, 95.4–97.7% on GPT2-XL). The difference reflects learning mechanisms: MLDD autonomously captures inter-feature covariance for scale adaptation, whereas IMSI relies on statistical convergence that benefits from larger samples.

### 6.4 ANALYSIS

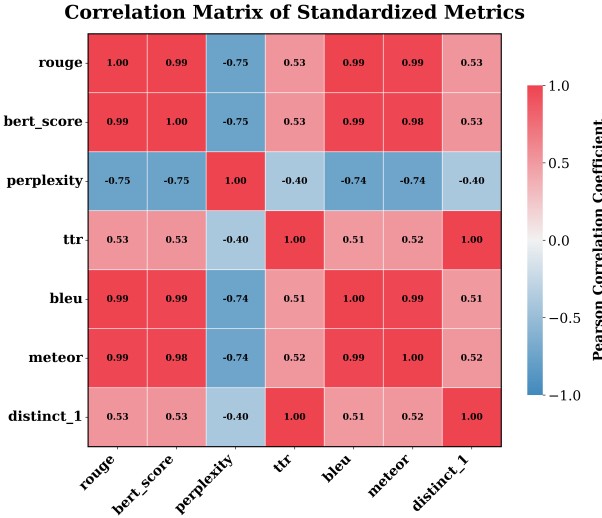

Figure 4: **Correlation heatmap of standardized metrics.** Pearson correlation among metrics across SLD, SCD, and QAD dimensions.

We first examine correlations within the proposed three-dimensional framework (SLD, SCD, QAD). As shown in Figure 4, BLEU and ROUGE are highly correlated ($r = 0.99$), confirming strong consistency in surface similarity. In SCD, BERTScore and Perplexity show a strong negative correlation

($r = -0.75$), reflecting that higher semantic similarity corresponds to reduced perplexity, a key indicator of memorized content. In QAD, TTR and Distinct-1 are perfectly correlated ($r = 1.0$), while moderately correlating with surface metrics ($r \approx 0.52$), suggesting partial lexical diversity preserved alongside surface replication. Overall, these results validate the framework's ability to capture complementary aspects of copyright memory.

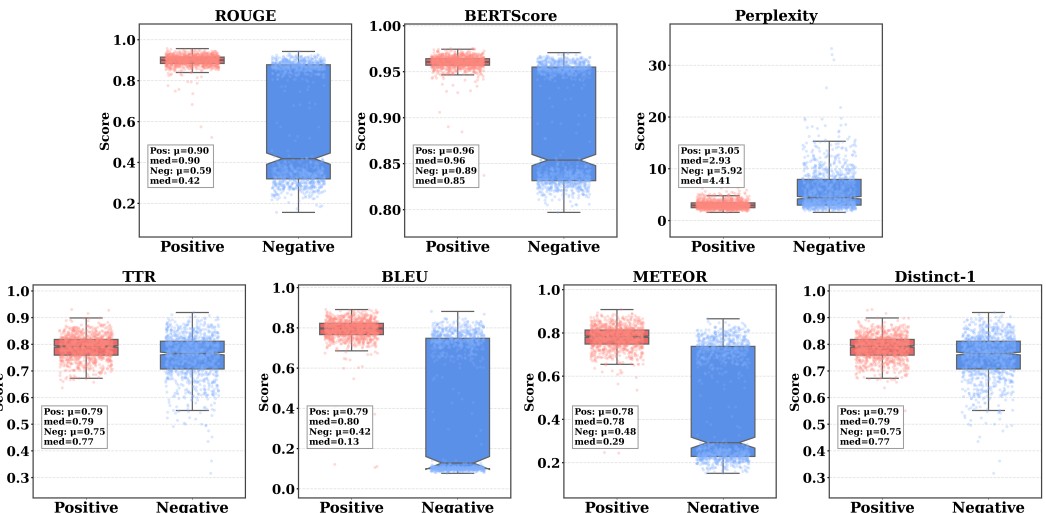

Figure 5: **Metric distributions for positive vs. negative samples.** Boxplots of seven core metrics under the tri-dimensional framework.

We then compare positive (C-FT) and negative (PD+C-NFT) samples (Figure 5). Positive samples show markedly higher ROUGE (+115%) and BLEU (+0.67), confirming surface-level memorization. In the semantic dimension, higher BERTScore and METEOR (+0.49) indicate strong semantic retention, while lower Perplexity signals increased fluency from fine-tuned memorization. In the quality dimension, slightly higher TTR and Distinct-1 suggest preserved lexical density but limited novelty due to source constraints. Together, these distributional gaps establish quantitative criteria for reliable copyright risk detection.

## 7 CONCLUSION

We presented **TRIDENT**, a three-dimensional framework for copyright infringement detection in LLMs that integrates surface, semantic, and quality features. By jointly modeling explicit replication and implicit paraphrasing behaviors, TRIDENT addresses the limitations of prior word-level or single-feature approaches. Our dual strategy—statistical significance testing for interpretability and learning-based detection for scalability—achieves near-perfect accuracy on GPT2-XL and DeepSeek-7B, reducing false positives by over 40% and boosting true positives to 99.7%. These results demonstrate that TRIDENT offers a reliable and generalizable solution for safeguarding copyright in large-scale language models. Future work will extend TRIDENT to multimodal scenarios and explore real-time deployment in practical systems.

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

## A   PSEUDOCODE

We implement two complementary detection methods within the TRIDENT framework. Algorithm 1 gives the pseudocode of the *Integrated-Metrics Statistical Inference (IMSI)* method, which applies statistical inference over multi-metric features to achieve interpretable detection. Algorithm 2 presents the *Machine Learning Direct Discrimination (MLDD)* method, which leverages feature-based classification for automated copyright discrimination.

---

**Algorithm 1** Integrated-Metrics Statistical Inference (IMSI)

---

**Symbols:** $\mathcal{M}_{\text{ref}}$: Reference model, $\mathcal{M}_{\text{tar}}$: Target model, $x_{\text{pre}}$: Text prefix, $\Phi$: metric function, $x_{\text{gen}}^{\text{ref}}/x_{\text{gen}}^{\text{tar}}$: Generated continuations, $\mathbf{m}_{\text{ref}}/\mathbf{m}_{\text{tar}}$: Metric vectors, $\mathbf{w}_c/\mathbf{w}_m$: PCA weight vectors, $s_{\text{ref}}/s_{\text{tar}}$: Composite scores, $\Delta_{\text{pub}}$: Public data baseline adjustment.
**Input:** Reference model $\mathcal{M}_{\text{ref}}$, target model $\mathcal{M}_{\text{tar}}$, sample $x$, $\Delta_{\text{pub}}$, $\tau$
**Output:** Copyright status $y \in \{0, 1\}$
Extract prefix: $x_{\text{pre}} \leftarrow x_{1:\lfloor \alpha n \rfloor}$
Generate: $x_{\text{gen}}^{\text{ref}} \leftarrow \mathcal{M}_{\text{ref}}(x_{\text{pre}})$, $x_{\text{gen}}^{\text{tar}} \leftarrow \mathcal{M}_{\text{tar}}(x_{\text{pre}})$
$\mathbf{m}_{\text{ref}} \leftarrow \Phi(x_{\text{gen}}^{\text{ref}}, x_{\text{ref}})$, $\mathbf{m}_{\text{tar}} \leftarrow \Phi(x_{\text{gen}}^{\text{tar}}, x_{\text{ref}})$
$\mathbf{w}_c \leftarrow \text{PCA}(\mathcal{F}_{\text{cat}})$, $\mathbf{w}_m \leftarrow \text{PCA}(\mathcal{C}_{\text{met}})$
**for** $i = 1 \rightarrow M$ **do**
    $w_i \leftarrow w_{c,i} \times w_{m,i}$
**end for**
$s_{\text{ref}} \leftarrow \sum_{i=1}^{M} w_i \cdot m_{\text{ref},i}$, $s_{\text{tar}} \leftarrow \sum_{i=1}^{M} w_i \cdot m_{\text{tar},i}$
$\Delta s \leftarrow s_{\text{tar}} - s_{\text{ref}}$
$\text{DID} \leftarrow \Delta s - \Delta_{\text{pub}}$
**if** $\text{DID} > \tau$ **then**
    **return** 1
**else**
    **return** 0
**end if**

---

# B ADDITIONAL EXPERIMENTAL RESULTS

## B.1 WEIGHT ANALYSIS

To resolve the homogeneity limitation in traditional equal-weighting methods regarding metric contributions, we implement hierarchical PCA to quantify discriminatory weights for three-dimensional metrics. As demonstrated in Figure 6, hierarchical PCA weighting distinctly reveals differential importance across evaluation dimensions for infringement discrimination, where the semantic dimension constitutes the primary discriminator at 39.3 % weighting, governing semantic-level infringement signal detection; the structural dimension contributes 35.6 % weighting to quantify surface replication intensity; and the quality dimension provides 25.1 % weighting for monitoring lexical diversity attenuation. This weighting distribution confirms semantic retention coupled with surface replication as core manifestations of copyright leakage. The weighting scheme objectively quantifies three-dimensional metric synergy through hierarchical PCA, demonstrating scientific superiority over naive equal-weighting strategies by autonomously amplifying combined semantic-structural contributions. Despite lower weighting, quality metrics uncover characteristic preservation of surface lexical diversity alongside innovation attenuation in infringing texts, which stratified weighting precisely extracts. This approach fundamentally resolves equal-weighting's limitation in neglecting metric contribution heterogeneity, enhancing interpretability and robustness in the resultant composite discrimination index.

## B.2 DID THRESHOLD ANALYSIS

To precisely disentangle structural memory induced by fine-tuning from base model fluctuations, we employed Difference in Differences (DID) methodology to quantify copyright memory effects, establishing causal inference through cross-model comparison. Figure 7 composite score distribution demonstrates, copyright-protected positive samples exhibit significantly higher median scores on the target model versus the reference model with markedly reduced dispersion, while negative samples maintain stable distributions across both models. This pattern confirms that parameter space reconstruction during fine-tuning establishes structural memory for copyrighted content. Figure 8 DID analysis effectively isolates base model fluctuations to further quantify this structural memory effect. Adjusted DID values for positive samples range from 1.25 to 2.11, with mean value of 1.68, showing complete separation from negative samples, ranging from negative 0.21 to 0.38 with a mean of 0.03. Applying Figure 9 Youden optimized threshold 0.9277, captures 98.7% of positive samples within the light-green detection zone. Figure 6 ROC curve verifies threshold efficacy, yielding an

**Algorithm 2** Machine Learning Direct Discrimination (MLDD)

---

**Symbols:** $D_{\text{train}}$: training set, $\mathbf{m}^i_{\text{ref}}/\mathbf{m}^i_{\text{tar}}$: metric vectors for sample $i$, $\Phi$: metric function, $\mathcal{C}$: classifier.

**Input:** Reference model $M_{\text{ref}}$, target model $M_{\text{tar}}$, training set $D_{\text{train}}$, test sample $x$

**Output:** Copyright status $y \in \{0, 1\}$

**for** each $x_i \in D_{\text{train}}$ **do**

    Extract prefix: $x^i_{\text{pre}} \leftarrow x_i^{1:\lfloor \alpha n \rfloor}$

    Generate: $x^{\text{ref},i}_{\text{gen}} \leftarrow M_{\text{ref}}(x^i_{\text{pre}})$

    Generate: $x^{\text{tar},i}_{\text{gen}} \leftarrow M_{\text{tar}}(x^i_{\text{pre}})$

    $\mathbf{m}^i_{\text{ref}} \leftarrow \Phi(x^{\text{ref},i}_{\text{gen}}, x_{\text{ref}})$

    $\mathbf{m}^i_{\text{tar}} \leftarrow \Phi(x^{\text{tar},i}_{\text{gen}}, x_{\text{ref}})$

    $\mathbf{f}^i \leftarrow [\mathbf{m}^i_{\text{ref}}; \mathbf{m}^i_{\text{tar}}]$    (Feature vector)

**end for**

$\mathcal{C} \leftarrow \text{TrainClassifier}(\{\mathbf{f}^i, y_i\}_{i=1}^{|D_{\text{train}}|})$

$\mathbf{f}^x \leftarrow [\mathbf{m}^x_{\text{ref}}; \mathbf{m}^x_{\text{tar}}]$    (Test sample features)

$y \leftarrow \mathcal{C}(\mathbf{f}^x)$

**return** $y$

---

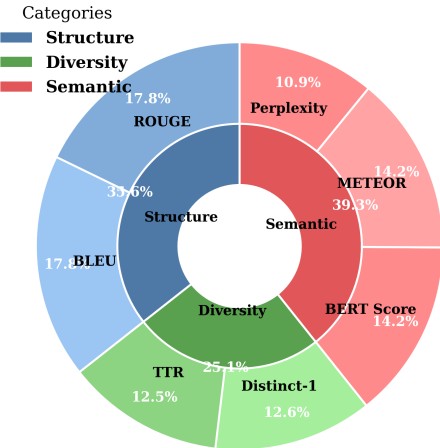

Figure 6: **Weight distribution of evaluation metrics.** Hierarchical PCA assigns 35.6% to surface features, 39.3% to semantic relevance, and 25.1% to quality. At the metric level, ROUGE and BLEU contribute 17.8% each, TTR and Distinct-1 12.5% each, BERTScore and METEOR 14.2% each, and Perplexity 10.9%.

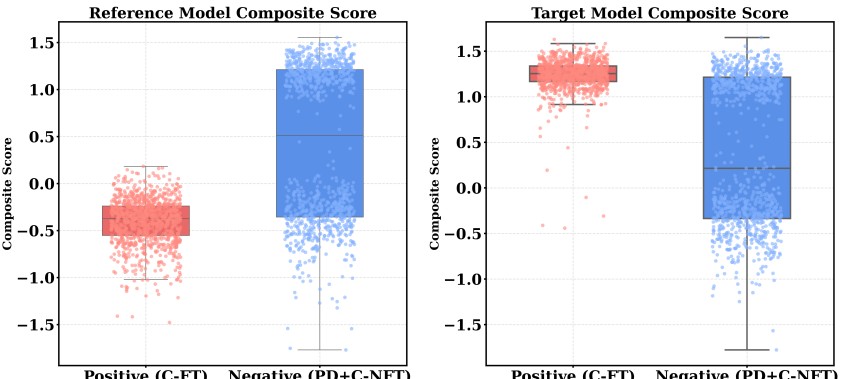

Figure 7: **Boxplots of Composite Score Distributions.** Comparison of positive and negative samples' composite scores between the reference model and the target model. Boxplots display medians and interquartile ranges, while scatter points reflect sample dispersion. Scores for the positive sample group on the target model were significantly higher than those on the reference model (upward shift marked in red), whereas the negative sample group exhibited relatively stable score distributions. These results suggest a specific memorization effect in the model toward copyrighted data.

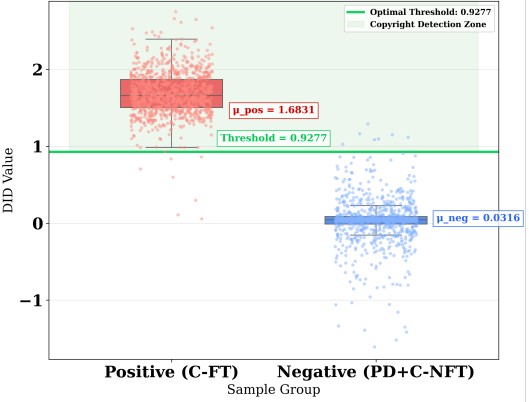

Figure 8: **Difference-in-Differences (DID) Distributions and Optimal Detection Threshold.** This figure visually presents the adjusted DID score distributions for copyrighted content (positive samples) and non-copyrighted content (negative samples) through boxplots and scatter plots. The solid green line indicates the Youden-optimal classification threshold derived from the ROC curve, with the light green area above it representing the copyright detection zone. Positive samples (marked in red) exhibit a significantly higher mean DID score (M=1.68) compared to negative samples' blue distribution (M=0.03), demonstrating the threshold's efficacy in distinguishing memorization effects of copyrighted content.

area under the curve AUC equal to 0.99 while balancing true positive rate 0.99 against false positive rate 0.01. This discriminative performance significantly surpasses random baselines

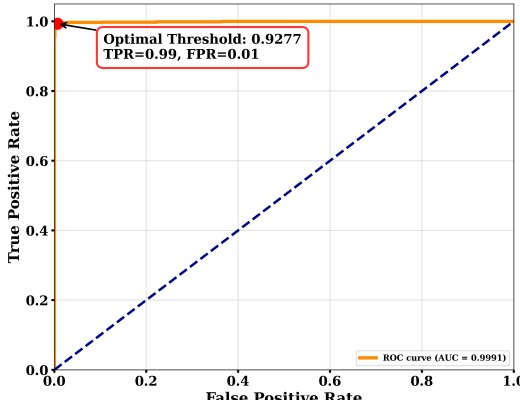

Figure 9: **ROC Curve at Optimal Threshold.** This figure illustrates the ROC curve performance of the copyright detection model. The area under the curve (AUC=0.99) indicates excellent discriminative ability. The red marker denotes the Youden-optimal threshold (0.9277), achieving a true positive rate (TPR) of 0.99 with a false positive rate (FPR) of 0.01. The solid orange line demonstrates substantial superiority over the random guessing baseline (blue dashed line).

