# OpenReview forum: "TRIDENT: Three-Dimensional Data Copyright Infringement Detection in LLMs"
_ICLR.cc/2026/Conference — Submitted to ICLR 2026_

### Official Review · Reviewer_9ZP5 · 2025-10-18

**Soundness:** 2
**Presentation:** 3
**Contribution:** 2
**Rating:** 2
**Confidence:** 3

**Summary:**

The paper introduces TRIDENT, a method for detecting whether an LLM has been exposed to copyright-protected content during training. Unlike prior approaches that rely on a single signal (e.g., word-overlap), TRIDENT combines three complementary dimensions: (1) surface-level similarity (word-overlap metrics), (2) semantic-level similarity (for capturing conceptual memorization), and (3) quality-level signals (e.g., lexical diversity) that reflect the creativity of generated text. These dimensions are used to extract features from a Reference model (not exposed to the copyrighted set) and a Target model (which is exposed to copyrighted data), and are then fed into two detectors: a trained Random Forest classifier and a more classical one using a Difference-in-Differences framework. On GPT-2-XL and DeepSeek-7B, both variants outperform recent detection baselines and achieve near-perfect AUC scores on a dataset of English novels collected by the authors.

**Strengths:**

- Detecting LLM training data is not entirely new, but it’s definitely an important and timely topic, especially regarding copyright-content detection. For that reason, trying to improve over current detectors, which are still not perfect, is a well-motivated goal.

- The idea of combining the three different dimensions is, to my knowledge, novel and I see it as the main contribution of the work.

- I also think the paper does mostly a good job in explaining the proposed method. I have some minor clarification questions, which I develop in the following sections.

**Weaknesses:**

- The method depends on a paired Reference vs. Target model comparison. In black-box models like GPT-5, even thought they sometimes allow fine-tuning, TRIDENT could only be used to test the presence of newly released books. With the current setup it remains unclear if it could be possible to construct such copyright-clean version of the reference model if we were interested in detecting an older book, like Harry Potter.

- Perplexity may be a good membership inference indicator but it also limits the scope of models that are able to run TRIDENT, due to requirement for token-level probability access.

- In Figure 5, the ROUGE-L distributions are almost perfect, and  Section 6.1 reports the models were fine-tuned for 10 epochs. After so much repeated exposure, memorization is unsurprising, but this setup does not seem comparable to how models are trained and deployed in practice.

**Questions:**

- Picking up on the weakness regarding generalization to black-box models: can TRIDENT be applied to a fully black-box LLM such as GPT-5?

- Is it necessary to include all the features in the RF classifier? As results show, some are highly correlated with each other, which may indicate redundancy.

- What specific benefit does PCA provide over simpler weight aggregation?

- Did the fine-tuning hurt the other model capabilities in any way? Since the models train for so many epochs on these books could they
have lost performance on other tasks? It would be great to see some experiments on this.

- What is the detection performance with different levels of exposure of the books? Per example, if the models only train for 1 epoch.

- I don’t see any references about possible release of code and the public-domain portion of the data. Will those be available?

---

### Official Review · Reviewer_1WM4 · 2025-10-30

**Soundness:** 2
**Presentation:** 2
**Contribution:** 2
**Rating:** 2
**Confidence:** 4

**Summary:**

The paper introduces a framework for detecting copyrighted content memorized by fine-tuned LLMs.
A reference model is first obtained by fine-tuning a base model on public data. For each candidate sequence suspected of being copyrighted, both the reference and target models are prompted with a prefix of the sequence, and their generated continuations are collected.
The approach then computes a range of similarity metrics between the generated and target sequences, capturing surface-level, semantic, and quality-based similarities.
Two inference methods are proposed to determine whether a sequence was seen during training:
- Integrated-Metrics Statistical Inference (IMSI): aggregates all similarity metrics, applies a difference-in-differences adjustment, and computes a final membership score using an optimal threshold.
- Machine Learning Direct Discrimination (MLDD): trains a random forest classifier on the metric features to predict membership directly.
The framework is evaluated against existing membership inference attacks (MIAs) for LLMs, with additional ablations on sequence and prefix length, and an analysis of how each similarity type contributes to detection performance.

**Strengths:**

- Detecting (copyright-protected) LLM training data is a timely and important problem
- The paper considers a broad range of similarity metrics for inferring membership

**Weaknesses:**

The main issues I find with this work is (a) the lack of clarity about the evaluation dataset, which makes it difficult to evaluate and ultimately trust the results (see below) and (b) the limited novelty, the method being an ensemble of existing similarity metrics.

Dataset:
- Previous work [1,2,3] has shown attacks against LLMs to be very sensitive to a distribution shift between member and non-member data and has proposed the use of model-less baselines, e.g., a simple Bag of Words classifier, to provide a measure of the distribution shift between member and non-member data. Adding a proper model-less baseline would provide a sound basis for the experimental setup and the reported results.
- Details are lacking about where the data is collected from or how the dataset is constructed.
- It is not clear what the MLDD model is trained vs evaluated on. Is the same data used for both? If so, the high performance metrics would seem misleading.
- IMSI is not explained clearly: In section 4.2, delta_pub is computed by taking the expectation of S(x) over records in D_pub. D_pub is never defined, nor do the authors explain what is contained in it. In 4.3, D_val is used to find an optimal threshold. D_val is also never defined.
- The method is only evaluated on one dataset (that is not public) and two models. This makes it difficult to know if the results will generalize to other datasets or are overfit to the specific dataset. Including the method’s performance on more (publicly accessible) datasets would strengthen the findings.

Novelty:
- The method does not seem particularly novel, being mostly an ensemble of existing similarity metrics without introducing a new modeling concept, theoretical insight, or attack formulation.
- The two proposed inference methods (IMSI and MLDD) are also relatively standard statistical and machine-learning approaches, applied without significant innovation.

Minor:
- TRIDENT is misspelled (“THRIDENT”) in Section 4 (line 224)
- New notations are introduced and never used again (e.g., H_0, H_1). This makes the paper difficult to read at times.
- The composite score weighting (Section 4.1) could be explained more clearly. I assume “class” and “category” refer to dimensions, but it would be helpful to state this clearly.

[1] Duan, Michael, et al. "Do Membership Inference Attacks Work on Large Language Models?." First Conference on Language Modeling. 2024
[2] Meeus, Matthieu, et al. "Sok: Membership inference attacks on llms are rushing nowhere (and how to fix it)." 2025 IEEE Conference on Secure and Trustworthy Machine Learning (SaTML). IEEE, 2025.
[3] Das, Debeshee, et al. "Blind baselines beat membership inference attacks for foundation models." 2025 IEEE Security and Privacy Workshops (SPW). IEEE, 2025.

**Questions:**

- Where is the dataset collected from?
- How many samples does the dataset contain, and how many of them are used for fine-tuning?
- What is the exact data split between training, validation, and test sets for the MLDD model? Was the model evaluated on unseen data? If not, why?
- What is contained in D_pub and D_val?
- In the captions of tables 1 and 2, it says “N=2000”. N is never defined, from the rest of the paper it seems to denote sample size, but in Section 6.1 (Experimental setup), it says that the dataset consists of 100 samples. Could you clarify what N denotes and how many samples are used for training/evaluating your method?

---

### Official Review · Reviewer_JS5V · 2025-11-01

**Soundness:** 2
**Presentation:** 2
**Contribution:** 1
**Rating:** 2
**Confidence:** 3

**Summary:**

This paper addresses copyright infringement in LLMs and proposes TRIDENT, built on the observation that copyrighted data in an LLM is reflected in surface features, semantic relevance, and generation quality. Specifically, the method comprises two detection methods: IMSI and MLDD. Extensive experiments demonstrate its superiority over other detection approaches.

**Strengths:**

1. The paper is well organized and easy to follow.
2. This paper integrates various metrics (e.g., BLEU and Perplexity) to build the detection approaches.

**Weaknesses:**

1. The technical contribution is a bit weak in this paper. The three-dimensional quantification framework incorporates well-studied metrics and classifies them into three dimensions: SLD, SCD, and QAD. To this end, the authors reorganize them for two detection approaches: IMSI and MLDD. I think authors should explicitly discuss the challenges when building these two approaches.
2. The method description is unclear. In Section 4.1, IMSI requires intra-class weights and inter-category weights, but there is no further discussion about how these weights are set. In the introduction, the authors state that "train a classifier for fully automated detection" for MLDD. However, I cannot see any details about training in Section 4.2.
3. The authors train reference and target models from a base model, which is impractical because these two models do not exist at the same time (i.e., one is trained by the copyrighted materials, while the other is trained by the non-copyrighted materials). Recent works on copyright protection use the pretrained base models (e.g., LLaMA and Qwen) and highlight their copyright infringement issues. How about when we want to report copyright infringement issues involving the proposed detection approaches for open-source models?

**Questions:**

**See Weaknesses**

---

### Official Review · Reviewer_9Yuv · 2025-11-02

**Soundness:** 2
**Presentation:** 2
**Contribution:** 2
**Rating:** 2
**Confidence:** 3

**Summary:**

The paper proposes a method for detecting LLM copyright infringement based on an aggregation of multiple metrics. The approach relies on an ensemble of metrics including BLEU, ROUGE, BERTScore, and Perplexity, as well as differences between a target model and a reference model. These features are then fed into a classifier to detect potential copyright infringement.

**Strengths:**

The topic of copyright infringement detection is important and relevant for the LLM community.

The idea of combining different metrics into an ensemble seems well-motivated for this type of problem.

**Weaknesses:**

The approach assumes access to both a reference model finetuned on non-copyrighted data and another model trained on both copyrighted and non-copyrighted data. This assumption does not seem very practical.

The experiments are limited to finetuning, where models are trained for 10 epochs. This seems quite high, and overfitting is likely at that point. In practice, finetuning would typically be done for fewer epochs. It would strengthen the paper if the authors could show results comparing their method and baselines when finetuning for only 1 epoch.

It is unclear whether the proposed method could be extended to pretraining data detection.

None of the baselines make use of reference information as the proposed method does, meaning they operate with less information. It would be useful to compare the proposed approach against a "reference"-type of baseline such as the one named "Small" from [1], but instead of using a smaller model as a reference as in [1] the authors can use the reference model from their paper.


References

[1] https://www.usenix.org/system/files/sec21-carlini-extracting.pdf

**Questions:**

If I understand this correctly, the novels are split into smaller chunks for detection. Is each chunk treated as a separate sample? How large is the detection set after splitting (i.e., the total number of samples, not just the number of novels)?

---

### Meta-Review · Area_Chair_s3wf · 2026-01-08

**Summary:**

The paper proposes a method for detecting LLM copyright infringement based on an aggregation of multiple metrics. The approach relies on an ensemble of metrics including BLEU, ROUGE, BERTScore, and Perplexity, as well as differences between a target model and a reference model. These features are then fed into a classifier to detect potential copyright infringement.

The reviewers raised a couple of common concerns regarding:
1) the practicality of availability of Target Model and Reference Model. This is critical since it serves as the backbone of the proposed method.
2) the clarity of the propose method as well as the evaluation datasets.
3) the innovation of the propose method are quite limited.
4) fair comparison against other baseline methods.

**Reviewer Concerns:**

The authors didn't submit a rebuttal to address these concerns.

**Reviewer Scores:**

It is unlikely that reviewers would change their scores, as no rebuttal was submitted.

---

### Decision · Program_Chairs · 2026-01-26

Reject